# Gestational Diabetes Mellitus and High Triglyceride Levels Mediate the Association between Pre-Pregnancy Overweight/Obesity and Macrosomia: A Prospective Cohort Study in Central China

**DOI:** 10.3390/nu14163347

**Published:** 2022-08-16

**Authors:** Xinli Song, Letao Chen, Senmao Zhang, Yiping Liu, Jianhui Wei, Tingting Wang, Jiabi Qin

**Affiliations:** 1Department of Epidemiology and Health Statistics, Xiangya School of Public Health, Central South University, Changsha 410078, China; 2National Health Committee Key Laboratory of Birth Defect for Research and Prevention, Hunan Provincial Maternal and Child Health Care Hospital, Changsha 410028, China; 3Guangdong Cardiovascular Institute, Guangdong Provincial People’s Hospital, Guangdong Academy of Medical Sciences, Guangzhou 510080, China; 4Hunan Provincial Key Laboratory of Clinical Epidemiology, Changsha 410078, China

**Keywords:** overweight, obesity, macrosomia, triglyceride, gestational diabetes mellitus, mediation analysis

## Abstract

The purpose of this study is to investigate whether the link between pre-pregnancy overweight/obesity and risk of macrosomia is mediated by both gestational diabetes mellitus (GDM) and high maternal triglyceride (mTG) levels. This prospective study finally included 29,415 singleton term pregnancies. The outcome of interest was macrosomia (≥4000 g). High mTG levels were denoted as values ≥90th percentile. GDM was diagnosed using a standard 75 g 2 h oral glucose tolerance test. The mediation analysis was conducted using log-binomial regression while controlling for maternal age, education, parity, gestational weight gain, gestational hypertension, smoking, drinking and infant sex. Overall, 15.9% of pregnant women were diagnosed with GDM, and 4.3% were macrosomia. Mediation analysis suggested that overweight had a total effect of 0.009 (95% CI, 0.006–0.013) on macrosomia, with a direct effect of 0.008 (95% CI, 0.004–0.012) and an indirect effect of 0.001 (95% CI, 0.001–0.002), with an estimated proportion of 11.1% mediated by GDM and high mTG levels together. Furthermore, we also discovered a total effect of obesity on macrosomia of 0.038 (95% CI, 0.030–0.047), consisting of a direct effect of 0.037 (95% CI, 0.028–0.045) and an indirect effect of 0.002 (95% CI, 0.001–0.002), with an estimated proportion of 5.3% mediated by GDM and high mTG levels combined. Both GDM and high mTG levels enhanced the risk of macrosomia independently and served as significant mediators in the relationship between pre-pregnancy overweight/obesity and macrosomia.

## 1. Introduction

Obesity and overweight among women commencing pregnancy represents a major clinical and public health issue in pregnancy care, with data suggesting a rising prevalence of body mass index (BMI) above 25 kg/m^2^ in Chinese women of childbearing age [1]. Increasing maternal BMI is a well-established determinant for a series of adverse pregnancy outcomes for both the mother and the infant, including maternal gestational diabetes mellitus (GDM) [2] and excessively elevated triglyceride levels [3], as well as their infant being born large for gestational age (LGA) and fetal macrosomia [4]. In recent years, it has been demonstrated that fetal overgrowth (macrosomia) is associated with an increased risk of lifelong consequences such as type 2 diabetes, obesity, high blood pressure, and cardiovascular problems [5,6], resulting in a surge of interest in investigating the factors affecting fetal overgrowth. Maternal nutrition and metabolism during pregnancy influences nutrients crossing the placenta contribute to fetal growth [7]. Maternal glucose is commonly regarded as the most important contributor in influencing birth weight in a continuous way [2]. In addition to glucose, maternal lipids, particularly triglycerides, are also important predictors of macrosomia and adiposity at birth [8]. A recent meta-analysis summarized the relationships between first-, second-, and third-trimester maternal triglyceride (mTG) levels (fasting, postprandial, or random) and fetal macrosomia among pregnancies of various races/ethnicities, compatible with the findings of most previous studies [4,9,10]. Freinkel contends that multiple maternal nutrients, such as glucose, lipids, and amino acids, interact to influence fetal growth and obesity, as well as diabetes later in life [11].

GDM and elevated mTG levels are both documented consequences of obesity and risk factors for the occurrence of fetal overgrowth, implying their potential mediating effects on this causal pathway. It has been demonstrated that both GDM [12] and high mTG levels [13] could play a role in mediating the relationship between pre-pregnancy BMI and macrosomia. Most pregnant women with GDM have an excess of elevated triglyceride levels at the same time [14]. As a result, it is sound to think about the combined mediating effects of GDM and high mTG levels in the connection between maternal pre-pregnancy overweight/obesity and the risk of macrosomia. Thus, from an epidemiological perspective, this study constructs a chain mediation model to understand how both GDM and high mTG levels act as mediators in the causal pathway of pre-pregnancy overweight/obesity on macrosomia. This study tests the following proposed assumptions based on this theoretical model: (i) the impact of pre-pregnancy overweight/obesity on GDM, high mTG levels, and macrosomia; and (ii) the mediating effect of GDM and high mTG levels in the impact of pre-pregnancy overweight/obesity on macrosomia; and (iii) GDM and high mTG levels have a chain mediating effect on the association of pre-pregnancy overweight/obesity with the risk of fetal macrosomia. With the support of theory, the objective of this prospective study was to determine the extent to which the association of pre-pregnancy overweight/obesity with the risk of macrosomia in singleton term pregnancies is mediated by both GDM and high mTG levels, in order to propose countermeasures and suggestions for improving perinatal outcomes.

## 2. Materials and Methods

### 2.1. Research Design and Study Population

We recruited first-trimester pregnant women who were aged equal to or more than 18 years old at the Hunan Provincial Maternal and Child Health Care Hospital between March 2013 and December 2019. We excluded pregnant women with pre-pregnancy diagnosis of type 1 or 2 diabetes and those with multiple pregnancies. Pregnant women without natural pregnancies and those who did not deliver at full term were also excluded. At the time of pregnancy registration, our specially trained researchers conducted face-to-face interviews with each enrolled woman, utilizing study-specific questionnaires to obtain their information, including age at the start of pregnancy, education, parity, and smoking and alcohol use during pregnancy. Maternal weight and height were measured while wearing light clothes without shoes to obtain pre-pregnancy body mass index (BMI). We followed all pregnant women until delivery. The increase from the pre-pregnancy weight to the weight at the last visit was regarded as gestational weight gain (kg). Clinical records were retrieved from the hospital’s electronic medical records and included maternal gestational hypertension and the sex and birth weight of the infants. Smoking during pregnancy was defined as smoking one or more cigarettes per day for at least three months prior to or during pregnancy. Alcohol use during pregnancy was defined as consuming alcohol one or more times per week before or during pregnancy.

After 28 weeks of gestation, fasting blood sampling collected in the morning was drawn to measure serum triglyceride levels after an overnight fast. The plasma triglyceride levels were detected by using a commercial enzymatic assay (Roche Diagnostics, Mannheim, Germany) and a Cobas c702 analyzer, with an inter-assay coefficient of variation <2.3%. In addition, between 24 and 28 weeks of gestation, all participants were required to undergo a standard 75 g 2 h oral glucose tolerance test (OGTT). The serum glucose levels were detected by applying an automated analyzer (Toshiba TBA-120FR, Tokyo, Japan). Following a diagnosis of GDM, women were given dietary and lifestyle advice, as well as medication such as insulin and metformin, and were encouraged to monitor their blood glucose at home and keep it within the recommended target ranges [15].

Each subject provided written informed consent before data collection. Ethical permission for the study was granted by the Ethics Committee for Clinical Research of Xiangya School of Public Health of Central South University (no. XYGW-2018-36). Additionally, we have registered this study in the Chinese Clinical Trial Registry Center (registration number: ChiCTR1800016635).

### 2.2. Exposure

Pre-pregnancy BMI was classified using the criteria for Chinese adults: underweight (<18.5 kg/m^2^), normal weight (18.5–23.9 kg/m^2^), overweight (24.0–27.9 kg/m^2^) and obesity (≥28.0 kg/m^2^) [16].

### 2.3. Outcome

As the outcome of interest, macrosomia was defined as a birth weight equal to or greater than 4000 g, regardless of sex [17]. Since gestational age was the major determinant of birth weight, this study focused on full-term births (37–41 completed weeks’ gestation). Gestational weeks were obtained using the last menstrual period data or the first accurate ultrasound examination if the menstruation was irregular [18].

### 2.4. Mediator

In this study, both GDM and high mTG levels were considered as mediators of interest on the relationship of maternal overweight/obesity with the risk of fetal macrosomia. The International Association of Diabetes and Pregnancy Study Group established the following cut-off values for the diagnosis of GDM: 5.1 mmol/L for fasting serum glucose, 10.0 mmol/L for 1-h serum glucose, or 8.5 mmol/L for 2-h serum glucose [19]. Based on previous studies [20,21], the cut-off point for mTG was considered to be the 90th percentile, and the cut-off value in this study was 5.67 mmol/L. The definition of a “high mTG level” was a value equal to or greater than the 90th percentile; conversely, a “low mTG level” was a value below the 90th percentile.

### 2.5. Covariates

Confounding factors in relation to pre-pregnancy BMI, GDM, mTG levels and macrosomia were considered as covariates in this study. Based on a study and review of literature [22,23,24,25,26,27,28], we selected several confounders as follows: education (*high school or less, some college, or bachelor’s+*), maternal age (*<25, 25–29, 30–34 or ≥35 years old*), gestational hypertension (*yes or no*), smoke (*yes or no*), drink (*yes or no*), parity (*primipara or multipara*), gestational weight gain (*<10, 10–20 or ≥20 kg*), and newborn sex (*male or female*).

### 2.6. Statistical Analysis

In our study sample of full-term pregnancies (*n* = 29,415), descriptive analysis was performed to show the distribution of the characteristics of pregnant women and infants group by high mTG levels, GDM and macrosomia. We estimated the prevalence of high mTG levels, GDM and macrosomia at each BMI group, and their 95% confidence intervals (95% CI). We carried out the mediation analysis, which was put forward by Baron and Kenny, based on the counterfactual framework for causal inference [29,30,31]. Based on their theory, the mediation effect was significant based on the premise that the associations were statistically significant for all path models. The mediation analysis divided the total effects (path D) into the direct effect (path d) and indirect effects (Figure 1). Since our study focused on the extent to which the two mediating factors worked, we further separated the indirect effects into those mediated by GDM (path a and path a′) and those mediated by high mTG levels (path b and path b′), as well as a chain-mediating effect of both GDM and high mTG levels (path c) (Figure 1). We used multiple log-binomial (log-linear) regression models to test the significance of the associations for paths a, a′, b, b′, c and D, after the adjustment for confounders. The strength of the association was assessed using relative risk ratios (RR) and their 95% CI [31]. Subsequently, the mediation analysis was performed by using the mediation package in R software to estimate the total effects, direct effects, and total indirect effects, which included a separate indirect effect mediated by GDM, a separate indirect effect mediated by high mTG levels, and a chain-mediating effect of a combination of GDM and high mTG levels. To estimate uncertainty, this study used a quasi-Bayesian Monte Carlo method with 10,000 simulations [32]. The mediated proportion was calculated by dividing the value of the indirect effect by the total effect, and it was used to assess the extent to which the association between pre-pregnancy overweight or obesity and macrosomia was co-mediated by GDM and high mTG levels. To assess consistency, sensitivity analysis was performed with the 85th percentile as the cut-off point for categorizing mTG levels. To denote statistical significance, a two-tailed *p* value of <0.05 was used. R version 3.6.2 was used for all statistical analyses (R Foundation for Statistical Computing, Vienna, Austria).

## 3. Result

### 3.1. Baseline Characteristics of the Study Population

A total of 40,650 subjects were enrolled. We excluded 11,235 participants for the following reasons: (i) artificial fertilization (*n* = 568, 1.4%); (ii) not term births (*n* = 3643, 9.0%); (iii) termination of pregnancy (*n* = 831, 2.0%); (iv) pregnant women with pre-existing diabetes (*n* = 240, 0.6%); (v) multiple pregnancies (*n* = 661, 1.6%); (vi) data missing (*n* = 1046; 2.6%) and loss to follow-up (*n* = 4246, 10.4%) (Figure A1). As a result, 29,415 singleton full-term pregnancies with pre-pregnancy in all BMI ranges were studied.

The distribution of characteristics of mothers and children in GDM, high mTG levels, and macrosomia is summarized in Table 1. Overall, 15.9% of pregnant women were given a diagnosis of GDM, 10.2% had high mTG levels in late pregnancy, and 4.3% of singleton term births were macrosomia. The majority of women (70.7%) had a normal BMI prior to pregnancy, with 12.6% overweight and 2.6% obese. Most women were between 25 and 34 years old, with 35.1% between the ages of 25 and 29 and 37.9% between the ages of 30 and 34. More than half of the pregnant women (51.7%) had a college degree, and only 1.0% and 1.6% had experience with smoking and alcohol, respectively. More than half of our participants (51.6%) were multipara, and more than half of the babies born to our participants (53.0%) were male. During pregnancy, nearly three-quarters of pregnant women (73.5%) gained weight within the normal range, and 3.2% were diagnosed with gestational hypertension.

### 3.2. Prevalence of High mTG Levels, GDM and Fetal Macrosomia in Each BMI Group

The prevalence of GDM, high mTG levels, and macrosomia based on each BMI group is shown in Table 2. Women with obesity prior to pregnancy had a higher prevalence of GDM and high mTG levels (31.4% and 14.7%, respectively), compared to overweight women (23.7% and 12.9%, respectively), women with normal BMI (15.4% and 10.1%, respectively) and underweight women (9.0% and 7.7%, respectively). Likewise, the prevalence of macrosomia in babies born to obese women was the highest (15.6%), higher than in babies born to overweight women (5.9%), women with normal BMI (3.9%) and underweight women (2.8%).

### 3.3. The Testing for Significance of Paths a, a′, b, b′, c, and D

The testing for significance of paths a, a′, b, b′, c, and D is shown in Table 3. The path a model was used to evaluate the effects of overweight/obesity on GDM, and when possible confounding factors were adjusted, positive and significant relationships were identified (overweight aRR = 1.56, 95% CI, 1.43–1.70; obesity aRR = 2.09, 95% CI, 1.78–2.45) (Table 3). The path b model was to test the significance of the effects of overweight/obesity on high mTG levels, and significant associations adjusted for confounding factors were found (overweight aRR = 1.20, 95% CI, 1.08–1.34; obesity aRR = 1.32, 95% CI, 1.07–1.62). The path a′ model was used to estimate the effects of GDM on macrosomia, and after adjusting for confounders, substantial associations were observed (overweight aRR = 1.66, 95% CI, 1.42–1.93; obesity aRR = 1.79, 95% CI, 1.53–2.10). The path b’ model was utilized to assess the association between high mTG levels and fetal macrosomia, and significant associations were found after the adjustment of confounders (overweight aRR = 2.93, 95% CI, 2.52–3.41; obesity aRR = 2.46, 95% CI, 2.08–2.90). In addition, path c was used to estimate the effects of pre-pregnancy overweight/obesity on macrosomia mediated by both GDM and high mTG levels, and we observed significant associations (overweight aRR = 1.89, 95% CI, 1.71–2.08; obesity aRR = 1.94, 95% CI, 1.74–2.15). Furthermore, the relationships between overweight/obesity and fetal macrosomia were assessed using the path D model. After adjustment for potential confounding factors, positively significant associations were revealed (overweight RR = 1.56, 95% CI, 1.33–1.83; obesity RR = 5.19, 95% CI, 4.17–6.46). Overall, on the basis of the fact that all path models were significant, mediation analysis could be further performed.

### 3.4. Mediation Analysis

The extent to which the association of overweight/obesity on fetal macrosomia was mediated by both GDM and high mTG levels is presented in Table 4, including total, direct and indirect effects. In the overweight group, mediation analysis suggested that the total effect of maternal overweight on fetal macrosomia was 0.009 (95% CI, 0.006–0.013; *p* < 0.001), which included a direct effect of 0.008 (95% CI, 0.004–0.012; *p* < 0.001) and an indirect effect of 0.001 (95% CI, 0.000–0.001; *p* < 0.001), and the estimated mediation proportion by GDM and high mTG levels together at 11.1%. In addition, in the obese group, mediation analysis showed that the total effect of maternal overweight on fetal macrosomia was 0.038 (95% CI, 0.030–0.047; *p* < 0.001), which included a direct effect of 0.037 (95% CI, 0.028–0.045; *p* < 0.001) and an indirect effect of 0.002 (95% CI, 0.001–0.002; *p* < 0.001), and the estimated mediation proportion by GDM and high mTG levels together at 5.3%.

Table 5 also showed the separate mediating effects of GDM and high mTG levels, as well as a chain-mediating effect on the relationship of overweight/obesity with fetal macrosomia. The path *X*_*M*_1__*Y* model indicated the mediating effect of GDM on the association between overweight/obesity and macrosomia, and significant associations in both overweight (*p* < 0.001) and obese women (*p* < 0.001) were obtained. The path *X*_*M*_2__*Y* model suggested that high mTG levels mediated the relationship of overweight/obesity with the occurrence of macrosomia, with significant associations observed in both overweight (*p* < 0.001) and obese (*p* < 0.01) women. The path *X*_*M*_1__*M*_2__*Y* model indicated the chain-mediating effect of a combination of GDM and high mTG levels on the relationship of overweight/obesity with the risk of macrosomia, and we found significant associations in both overweight (*p* < 0.01) and obese women (*p* < 0.05).

To ensure consistency, we repeated the mediation analysis when mTG levels were cut off at the 85th percentile. We found that the results for total, direct, and indirect effects were similar to those described with the cutoff at the 90th percentile in both overweight and obese women (Table A1).

## 4. Discussion

It is well documented that maternal overweight/obesity as the strongest risk factor increases the risk of immediate consequences such as GDM and an excess of elevated mTG level, which subsequently carries a higher risk of fetal macrosomia. The physiological changes in insulin and lipids, particularly triglycerides, are exaggerated in women with GDM, and studies on circulating lipid patterns in GDM versus normal pregnancy found higher triglyceride levels in women with GDM across all trimesters of pregnancy [33]. GDM and high mTG levels coexist commonly, and they share some metabolic characteristics. Albeit the fact that GDM and high mTG levels interacts dynamically as pregnancy progresses, previous research only focused on their independent mediating effects in mediating the association between pre-pregnancy overweight/obesity and macrosomia [12,13]. With this background in mind, we sought to construct a chain mediation model in this hospital-based cohort study to better understand how both GDM and high mTG levels in late pregnancy act as mediators in the causal pathway of pre-pregnancy overweight/obesity on macrosomia among singleton full-term pregnancies from an epidemiological perspective.

It was thought that glucose was the primary source of energy for fetal growth, and altered maternal glucose homeostasis was a well-established risk factor for excessive fetal growth. In women with GDM, the excessive shunting of nutrients to the fetus and the acceleration of the fetal growth trajectory increased the risk of macrosomia [34]. Pedersen suggested that the transfer of excess maternal glucose stimulated the fetal islets, raising insulin levels and, as a consequence, increasing fetal glucose consumption and the risk of macrosomia [35]. Our findings suggested that GDM might play a mediating role in the association between maternal overweight/obesity and the risk of fetal macrosomia, independent of mTG levels during pregnancy and gestational weight gain, basically in line with our previously published findings [36]. In a cross-sectional study of singleton full-term American pregnancies, Kondracki and colleagues [12] discovered a mediating effect of GDM on the association of overweight/obesity and LGA births, which was in line with our observations. Additionally, regardless of birth weight, women with GDM were predisposed to having babies with adiposity (newborn %fat), a surrogate for newborn body composition, and macrosomic fetuses were always accompanied by some degree of increased body fat and even adiposity [37,38]. Babu and colleagues conducted a cohort study in South India and observed that the association of maternal obesity and fetal adiposity was partially mediated by GDM, which supported our findings [39]. Notably, our findings, as well as those of Kondracki et al. and Babu et al., were all based on the absence of follow-up glycemic control after the diagnosis of GDM and thus represented the significant mediating effect of GDM without good glycemic control. It was worthwhile to consider whether the mediating effect of well-treated GDM would be insignificant if women with GDM accepted optimal blood glucose control with blood glucose levels in the target range. Poprzeczny and colleagues [40] found an insignificant mediating effect of treated GDM in the association between increased maternal BMI and fetal adiposity in a randomized trial of diabetic women with the standard care group and the well-treated blood glucose group, which further provided evidence for our findings.

Recent efforts have yielded conclusive results in terms of interventions aimed at optimizing blood glucose levels in reducing the risk of fetal overgrowth [41,42]. In addition, growing evidence suggests that an early diagnosis and treatment of GDM in high-risk pregnant women would be more beneficial, whereas efforts to intervene after 34 weeks of gestation or later appear to be futile. Li and colleagues modeled fetal growth trajectories and found that the onset of GDM-related fetal overgrowth (e.g., the initial acceleration of fetal growth and fat mass accretion) could be detected at 20 weeks of gestation [43]. Sovio and colleagues reported that the differences in fetal size (e.g., fetal abdominal circumference and head circumference) between women with GDM and those without GDM became significant between 20 and 28 weeks of gestation [44]. Furthermore, early diagnosis of GDM (before 20 weeks of gestation) and prompt treatment have been shown to prevent the onset of fetal overgrowth in high-risk pregnancies [43]. Thereby, from a cost-effectiveness standpoint, an approach including earlier GDM screening based on personal risk profile may be preferable [45].

The influence of maternal lipids on birth weight was regarded as crucial as the well-established effect of glucose [46]. Our findings also revealed that high mTG levels in late pregnancy acted as a mediator in the relationship between maternal overweight/obesity and the risk of fetal macrosomia, independent of the mediating effect of GDM, and several potentially confounding factors such as age, education, parity, smoking, drinking, gestational weight gain, and gestational hypertension. Lu and colleagues conducted a prospective cohort study in the Chinese population and reported a significant mediating effect of TG levels in the association between pre-pregnancy BMI and macrosomia with BMI considered as a continuous variable [13], which supported our findings. Our previously published research indicated that high mTG levels could mediated the risk of macrosomia in relation to maternal pre-pregnancy overweight/obesity in singleton term non-diabetic pregnancies [47]. Maternal lipid metabolism changes, such as early lipid accumulation in maternal tissue and the development of hyperlipidemia in the second half of pregnancy, were considered normal physiology during pregnancy [48]. Maternal lipids, such as triglycerides, were important substrates in addition to glucose for fetal fat accretion, especially in late pregnancy when adipogenesis accelerated [8]. Although lipids cannot cross the placenta directly, placental trophoblasts may transport triglyceride hydrolysates, such as free fatty acids, via specific fatty acid binding/transport proteins [8]. The increased transport of specific fatty acids to the placenta and fetus caused metabolic adaptations, which contributed to macrosomia [49]. Olmos and colleagues concluded from a prospective study that, despite optimal maternal glucose control throughout pregnancy, mTG levels were still responsible for macrosomic babies to some extent [50], further supporting our observation.

Furthermore, we discovered a chain-mediating effect of high mTG levels and GDM in the causal pathway between maternal pre-pregnancy overweight/obesity and the risk of fetal macrosomia. Normal gestational metabolism, regulated by multiply hormones, was accompanied by a physiological increase in glucose, insulin resistance and insulin levels, as well as serum lipids such as triglycerides and free fatty acids, which resembled a “metabolic syndrome” as outlined beyond pregnancy [8]. It was indicated that women with GDM were at a significantly higher risk of developing high triglyceride levels than women without insulin resistance, and this was consistent throughout the whole pregnancy [14]. Ryckman and colleagues discovered that pre-pregnancy BMI accounted for 11% of the heterogeneity in the relationship between mTG levels and GDM. It was uncertain whether hypertriglyceridemia took place only later in pregnancy after the onset of GDM or if the dyslipidemia occurred prior to the onset of insulin resistance. Hypertriglyceridemia, rather than hypercholesterolemia, and free fatty acids were the cause of worsening insulin resistance [51]; however, the combination of increased fat consumption and worsening insulin resistance resulted in lipoprotein lipase deficiency, which contributed to early differences in mTG levels between women with and without GDM [14]. Further research was warranted to uncover the mechanisms of the relationship between maternal pre-pregnancy overweight/obesity, GDM, high mTG levels, and fetal macrosomia.

Glucose and TG were both the major energy substrates for fetal size growth, but prior literature only takes one into account as a mediator. The principal strength and novelty of this study was that it was the first time considering the mediating effect of both GDM and high mTG levels associated with overweight/obesity and macrosomia. The primary weaknesses of this study are as follows: (i) our subjects came from a single city region, and they represented only a subset of the population in central China; replications are still needed to further confirm our observations; (ii) the offspring’s glucose and triglyceride profiles were not collected, which could help explain the proposed relationship between maternal GDM, high mTG levels, and fetal macrosomia; (iii) this study did not follow up on the interventions given to women with GDM, such as proper weight gain, a healthy diet and physical activity, regular glucose monitoring, and medication therapy; additionally, the glucose control status was not recorded. This study was a preliminary investigation on this topic, and future research is warranted.

## 5. Conclusions

From an epidemiologic perspective, we conclude that both GDM and high mTG levels are significant mediators in the relationship between pre-pregnancy overweight/obesity and macrosomia.

## Figures and Tables

**Figure 1 nutrients-14-03347-f001:**
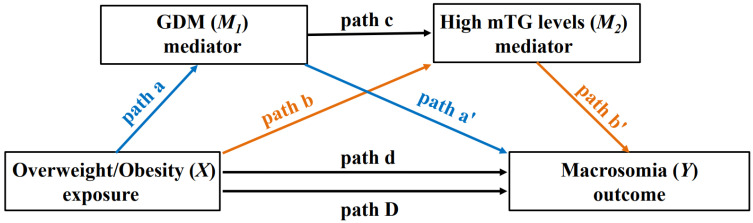
The illustration of the total effect (path D), indirect effect (path a, a′, b, b′ and c) and direct effect (path d) on the relationship of the exposure (*X*) with outcome (*Y*) via the mediator (*M*_1_ and *M*_2_). Abbreviations: GDM, gestational diabetes mellitus; mTG, maternal triglyceride.

**Table 1 nutrients-14-03347-t001:** Maternal and newborn characteristics were distributed in the high mTG levels, GDM and macrosomia groups.

Characteristics	Total Births *n* (%)	GDM *n* (%)	High mTG *n* (%)	Macrosomia *n* (%)
	*n* = 29,415	4685 (15.9)	3008 (10.2)	1271 (4.3)
Pre-pregnancy BMI (kg/m^2^)				
Underweight (<18.5)	4153 (14.1)	372 (7.9)	318 (10.6)	117 (9.2)
Normal (18.5–23.9)	20,785 (70.7)	3192 (68.1)	2098 (69.7)	816 (64.2)
Overweight (24.0–27.9)	3699 (12.6)	877 (18.7)	478 (15.9)	217 (17.1)
Obese (≥28.0)	778 (2.6)	244 (5.2)	114 (3.8)	121 (9.5)
Age at pregnancy onset (year)				
<25	1451 (4.9)	100 (2.1)	142 (4.7)	68 (5.4)
25–29	10,335 (35.1)	1232 (26.3)	839 (27.9)	445 (35.0)
30–34	11,145 (37.9)	1897 (40.5)	1123 (37.3)	490 (38.6)
≥35	6484 (22.0)	1456 (31.1)	904 (30.1)	268 (21.1)
Education				
High school or less	10,179 (34.6)	1554 (33.2)	1108 (36.8)	436 (34.3)
Some college	15,202 (51.7)	2515 (53.7)	1579 (52.5)	668 (52.6)
Bachelor’s or higher	4034 (13.7)	616 (13.1)	321 (10.7)	167 (13.1)
Smoke				
No	29,124 (99.0)	4648 (99.2)	2981 (99.1)	1257 (98.9)
Yes	291 (1.0)	37 (0.8)	27 (0.9)	14 (1.1)
Drink				
No	28,958 (98.4)	4605 (98.3)	2950 (98.1)	1253 (98.6)
Yes	457 (1.6)	80 (1.7)	58 (1.9)	18 (1.4)
Parity				
Primipara	14,236 (48.4)	2148 (45.8)	1388 (46.1)	600 (47.2)
Multipara	15,179 (51.6)	2537 (54.2)	1620 (53.9)	671 (52.8)
Infant sex				
Male	15,582 (53.0)	2398 (51.2)	1609 (53.5)	822 (64.7)
Female	13,833 (47.0)	2287 (48.8)	1399 (46.5)	449 (35.3)
Gestational weight gain (kg)				
<10	4478 (15.2)	1105 (23.6)	506 (16.8)	120 (9.4)
10–20	21,630 (73.5)	3277 (69.9)	2134 (70.9)	907 (71.4)
≥20	3307 (11.2)	303 (6.5)	368 (12.2)	244 (19.2)
Gestational hypertension				
No	28,473 (96.8)	4523 (96.5)	2916 (96.9)	1243 (97.8)
Yes	942 (3.2)	162 (3.5)	92 (3.1)	28 (2.2)

Abbreviations: GDM, gestational diabetes mellitus; mTG, maternal triglyceride; BMI, body mass index.

**Table 2 nutrients-14-03347-t002:** The prevalence of high mTG levels, GDM and macrosomia in each BMI group.

Category	GDM % (95% CI)	High mTG % (95% CI)	Macrosomia % (95% CI)
Underweight (<18.5)	9.0 (0.81–0.98)	7.7 (6.8–8.5)	2.8 (2.3–3.3)
Normal (18.5–23.9)	15.4 (14.9–15.8)	10.1 (9.7–10.5)	3.9 (3.7–4.2)
Overweight (24.0–27.9)	23.7 (22.3–25.1)	12.9 (11.8–14.0)	5.9 (5.1–6.6)
Obese (≥28.0)	31.4 (28.1–34.6)	14.7 (12.2–17.1)	15.6 (13.0–18.1)

Abbreviations: GDM, gestational diabetes mellitus; mTG, maternal triglyceride; 95% CI, 95% confidence interval.

**Table 3 nutrients-14-03347-t003:** The testing for significance of paths a, a′, b, b′, c, and D.

Category	Path a aRR (95%CI) ^a^	Path a′ aRR (95%CI) ^b^	Path b aRR (95%CI) ^c^	Path b′ aRR (95%CI) ^d^	Path c aRR (95%CI) ^e^	Path D aRR (95%CI) ^f^
Overweight	1.56 (1.43–1.70)	1.66 (1.42–1.93)	1.20 (1.08–1.34)	2.93 (2.52–3.41)	1.89 (1.71–2.08)	1.56 (1.33–1.83)
Obese	2.09 (1.78–2.45)	1.79 (1.53–2.10)	1.32 (1.07–1.62)	2.46 (2.08–2.90)	1.94 (1.74–2.15)	5.19 (4.17–6.46)

Note: path a (mediator model for GDM): the relationship of overweight/obesity with GDM; path a’ (outcome model for GDM): the relationship of GDM with fetal macrosomia; path b (mediator model for high mTG levels): the relationship of overweight/obesity with high mTG levels; path b’ (outcome model for high mTG levels): the relationship of high mTG levels with fetal macrosomia; path c (mediator model): the relationship of GDM with high mTG levels; path D: the relationship of overweight/obesity with fetal macrosomia. Abbreviations: GDM, gestational diabetes mellitus; mTG, maternal triglyceride; 95% CI, 95% confidence interval; aRR, adjusted relative risk ratio. ^a^ Adjusted for maternal education and age, drinking, smoking, gestational weight gain, gestational hypertension, mTG levels, newborn sex and parity. ^b^ Adjusted for overweight/obesity, maternal education and age, drinking, smoking, gestational weight gain, gestational hypertension, mTG levels, newborn sex and parity. ^c^ Adjusted for maternal education and age, drinking, smoking, gestational weight gain, gestational hypertension, GDM, newborn sex and parity. ^d^ Adjusted for overweight/obesity, maternal education and age, drinking, smoking, gestational weight gain, gestational hypertension, GDM, newborn sex and parity. ^e^ Adjusted for overweight/obesity, maternal education and age, drinking, smoking, gestational weight gain, gestational hypertension, newborn sex and parity. ^f^ Adjusted for maternal education and age, drinking, smoking, parity, infant sex, gestational weight gain, gestational hypertension, GDM, mTG levels, newborn sex and parity.

**Table 4 nutrients-14-03347-t004:** Mediation effects of high mTG levels in combination with GDM on the relationship of overweight/obesity with macrosomia.

Category	Total Effect (95% CI)	Direct Effect (95% CI)	Indirect Effect (95% CI)	Mediated Proportion, %
Overweight	0.009 (0.006–0.013) ***	0.008 (0.004–0.012) ***	0.001 (0.001–0.002) ***	11.1
Obese	0.038 (0.030–0.047) ***	0.037 (0.028–0.045) ***	0.002 (0.001–0.002) ***	5.3

Note: Adjusted for maternal age and education, drinking, smoking, gestational weight gain, gestational hypertension, newborn sex and parity; *** *p* < 0.001. Abbreviations: GDM, gestational diabetes mellitus; mTG, maternal triglyceride; 95% CI, 95% confidence interval.

**Table 5 nutrients-14-03347-t005:** The separate mediating effect of GDM, high mTG levels and their combination on the relationship of overweight/obesity with the risk of macrosomia.

Category	*X*_*M*_1__*Y* (95% CI)	*X*_*M*_2__*Y* (95% CI)	*X*_*M*_1__*M*_2__*Y* (95% CI)
Overweight	0.001 (0.000, 0.001) ***	0.001 (0.000, 0.001) ***	0.000 (0.000, 0.000) **
Obese	0.001 (0.001, 0.002) ***	0.001 (0.000, 0.001) **	0.000 (0.000, 0.000) *

Note: *X*, pre-pregnancy overweight/obesity; *M*_1_, GDM; *M*_2_, high mTG levels; *Y*, macrosomia; *X*_*M*_1__*Y*: the mediation effect of GDM on the relationship of overweight/obesity with the risk of macrosomia; *X*_*M*_2__*Y*: the mediation effect of high mTG levels on the association between pre-pregnancy overweight/obesity and fetal macrosomia; *X*_*M*_1__*M*_2__*Y*: the chain mediating effect of a combination of GDM and high mTG levels on the relationship of overweight/obesity with the risk of macrosomia. Adjusted for maternal age and education, drinking, smoking, gestational weight gain, gestational hypertension, newborn sex and parity; * *p* < 0.05, ** *p* < 0.01, *** *p* < 0.001. Abbreviations: GDM, gestational diabetes mellitus; mTG, maternal triglyceride; 95% CI, 95% confidence interval.

## Data Availability

The data presented in this study are available on request from the corresponding author.

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
