# Peer review of "Gestational Diabetes Mellitus and High Triglyceride Levels Mediate the Association between Pre-Pregnancy Overweight/Obesity and Macrosomia: A Prospective Cohort Study in Central China"

_nutrients, 2022, doi:10.3390/nu14163347_

Round 1

Reviewer 1 Report

The authors have reported that the gestational diabetes mellitus and high triglyceride levels mediate the association between pre-pregnancy over weight/obesity and macrosomia. I agree with the authors that the investigated problem is important because fetal overgrowth (macrosomia) is associated with an increased risk of type 2 diabetes, obesity, high blood pressure, cardiovascular diseases and non-alcoholic fatty liver disease in children. I agree that glucose and TG are both the major energy substrates for fetal growth and the novelty of this study was to consider the mediating effect of both gestational diabetes mellitus and high TG levels associated with overweight/obesity and macrosomia. The research presented in this manuscript are probably a continuation of previous research published in Nutrients [1,2]. I think it would be good to refer to them in the discussion:

1.      Song X., et al. Pre-Pregnancy Body Mass Index and Risk of Macrosomia and Large for Gestational Age Births with Gestational Diabetes Mellitus as a Mediator: A Prospective Cohort Study in Central China. Nutrients 2022;14:1072. doi: 10.3390/nu14051072.

2.      Song X., et al. High Maternal Triglyceride Levels Mediate the Association between Pre-Pregnancy Overweight/Obesity and Macrosomia among Singleton Term Non-Diabetic Pregnancies: A Prospective Cohort Study in Central China. Nutrients 2022, 14: 2075. https://doi.org/10.3390/nu14102075.

Author Response

Responds to the Reviewer #1

Comment: The authors have reported that the gestational diabetes mellitus and high triglyceride levels mediate the association between pre-pregnancy over weight/obesity and macrosomia. I agree with the authors that the investigated problem is important because fetal overgrowth (macrosomia) is associated with an increased risk of type 2 diabetes, obesity, high blood pressure, cardiovascular diseases and non-alcoholic fatty liver disease in children. I agree that glucose and TG are both the major energy substrates for fetal growth and the novelty of this study was to consider the mediating effect of both gestational diabetes mellitus and high TG levels associated with overweight/obesity and macrosomia. The research presented in this manuscript are probably a continuation of previous research published in Nutrients [1,2]. I think it would be good to refer to them in the discussion:

  1. Song X., et al. Pre-Pregnancy Body Mass Index and Risk of Macrosomia and Large for Gestational Age Births with Gestational Diabetes Mellitus as a Mediator: A Prospective Cohort Study in Central China. Nutrients 2022;14:1072. doi: 10.3390/nu14051072.
  2. Song X., et al. High Maternal Triglyceride Levels Mediate the Association between Pre-Pregnancy Overweight/Obesity and Macrosomia among Singleton Term Non-Diabetic Pregnancies: A Prospective Cohort Study in Central China. Nutrients 2022, 14: 2075. https://doi.org/10.3390/nu14102075.

Response: We sincerely appreciate your careful review and this kind suggestion, and we have cited the two references in the Discussion section as follows:

“Our findings suggested that GDM might play a mediating role in the association between maternal overweight/obesity and the risk of fetal macrosomia, independent of mTG levels during pregnancy and gestational weight gain, basically in line with our previously published findings[36].” (Page 9, Line 325-328)

“Our previously published research indicated that high mTG levels could mediated the risk of macrosomia in relation to maternal pre-pregnancy overweight/obesity in singleton term non-diabetic pregnancies[47].” (Page 9, Line 359-361)

Reviewer 2 Report

The authors aim to investigate the roles of maternal triglyceride concentrations and maternal GDM in the pathogenesis of fetal macrosomia in a large series.  They then extend these data to a model of the causes of fetal macrosomia.  The abstract is sound.  The introduction possibly too long and part could be incorporated into the discussion.  The methods could include a bit more for a non-Chinese audience.  The results are well described.  The discussion could consider moving the hypotheses to a different paper.  Overall the English is fair.

Specifically

19. I think it would be improved with more of the actual data in the abstract.  This is an editorial decision

GDM was diagnosed

20. interpreted how?

29. Previous work showing association maternal TG and birthweight.  Some of those papers include

Knopp Diabetes care  15:1605, Nolan Diabetes care  18:1550, Kitajima OG 97:776, Di Cianni Diabetic medicine 22:21, Clausen EJE 153:887, Ong Diabetes care  31:2193, Schaeffer DC 31:1858, Kulkarni Diabetes care  36:2706

The lancet paper is very useful on predictors of macrosomia worldwide.  Koyanagi 2013 Lancet 381:476-83

78.  Fasting TG is of interest.  Some have suggested the ability of insulin to suppress [TG] in a post glucose load has more predictive power.  Why fasted TG?  Please put actual TG values.

106. Some data of gestational age at delivery and consider the data on Large for gestational age.  Was gestational age confirmed by an early ultrasound about 14weeks?

106.  Macrosomia >4.5kg aswell?

106. Why was 4kg used as a cutoff.  Either mothers are being delivered early or 4kg is too high a definition in this population.  One considers this as only  % had a weight >4kg (not 10%) 

162 Table 1.  Please analyse the fasting and 2hr [glucose] and [TG] as continuous variables.  Our cutoffs for abnormal are arbitrary.  This could be done in addition to the method in the paper

211 percent of what?  Sorry These do not add to 100% even with non GDM, Normal low TG, macrosomia accounted for

Positive predictive value of BMI >28,  TG>what value 

Do you have data on actual outcomes eg shoulder dystocia

312 sorry I don’t understand the the English. Presumably they are stating It was thought that Glucose was the…

Some have suggested that Maternal insulin resistance is important for maternal hyperglycaemia (with increased amino acid flux to fetus driving fetal hyperinsulinaemia in addition to glucose which drives fatal hyperinsulinaemia AND contributes to the calories) and hypertriglycerideaemia (driving calories to fetus with glucose)

Both glucose and TG concentrations appear important

Author Response

Responds to the Reviewer #2

Comment 1: The authors aim to investigate the roles of maternal triglyceride concentrations and maternal GDM in the pathogenesis of fetal macrosomia in a large series. They then extend these data to a model of the causes of fetal macrosomia. The abstract is sound. The introduction possibly too long and part could be incorporated into the discussion. The methods could include a bit more for a non-Chinese audience. The results are well described. The discussion could consider moving the hypotheses to a different paper. Overall the English is fair.

Response 1: We sincerely appreciate your careful review and these constructive suggestions!

For the Introduction section, we have moved one sentence from this Introduction to the Discussion section as follows:

“Pedersen suggested that the transfer of excess maternal glucose stimulated the fetal islets, raising insulin levels and, as a consequence, increasing fetal glucose consumption and the risk of macrosomia[8].” (Page 9, Line 324-325)

For the Methods section, our study is indeed limited to the Chinese population, as described in the Limitations part as follows:

“our subjects came from a single city region, and they represented only a subset of the population in central China; replications were still needed to further confirm our observations;” (Page 10, Line 391-392)

Comment 2: 19. I think it would be improved with more of the actual data in the abstract. This is an editorial decision

GDM was diagnosed

  1. interpreted how?

Response 2: We sincerely thank you for these specific comments about the Abstract section. Based on your kind suggestions, we have rewritten the Abstract section as follows:

“Mediation analysis suggested that overweight had a total effect of 0.009 (95% CI, 0.006-0.013) on macrosomia, with a direct effect of 0.008 (95% CI, 0.004-0.012) and an indirect effect of 0.001 (95% CI, 0.001-0.002), with an estimated proportion of 11.1% mediated by GDM and high mTG levels together. Furthermore, we also discovered a total effect of obesity on macrosomia of 0.038 (95% CI, 0.030-0.047), consisting of a direct effect of 0.037 (95% CI, 0.028-0.045) and an indirect effect of 0.002 (95% CI, 0.001-0.002), with an estimated proportion of 5.3% mediated by GDM and high mTG levels combined.” (Page 1, Line 22-28)

Comment 3: 29. Previous work showing association maternal TG and birthweight. Some of those papers include

Knopp Diabetes care  15:1605, Nolan Diabetes care  18:1550, Kitajima OG 97:776, Di Cianni Diabetic medicine 22:21, Clausen EJE 153:887, Ong Diabetes care  31:2193, Schaeffer DC 31:1858, Kulkarni Diabetes care  36:2706

The lancet paper is very useful on predictors of macrosomia worldwide.  Koyanagi 2013 Lancet 381:476-83

Response 3: We sincerely thank you for this comment. We have cited your recommended references in the Introduction and Discussion section as follows:

“A recent meta-analysis summarized the relationships between first-, second-, and third-trimester maternal triglycerides (mTG) levels (fasting, postprandial, or random) and fetal macrosomia among pregnancies of various races/ethnicities, compatible with the findings of most previous studies[4, 10, 11].” (Page 2, Line 46-49)

“The influence of maternal lipids on birth weight was regarded as crucial as the well-established effect of glucose[46].” (Page 9, Line 353-354)

“It has been demonstrated from an epidemiologic standpoint that any high mTG levels detected in the first, second or third trimester were associated with an increased risk of having a macrosomic newborn, and this topic was confirmed in different races[4, 10, 11].” (Page 10, Line 367-370)

Comment 4: 78.  Fasting TG is of interest.  Some have suggested the ability of insulin to suppress [TG] in a post glucose load has more predictive power. Why fasted TG? Please put actual TG values.

  1. Some data of gestational age at delivery and consider the data on Large for gestational age.  Was gestational age confirmed by an early ultrasound about 14weeks?
  2. Macrosomia >4.5kg aswell?
  3. Why was 4kg used as a cutoff.  Either mothers are being delivered early or 4kg is too high a definition in this population.  One considers this as only  % had a weight >4kg (not 10%)

Response 4: We sincerely thank you for these comments.

The reasons for choosing fasting TG was following. On the one hand, four large prospective studies observed that maximal mean changes were +0.3 mmol/L for triglycerides, -0.2 mmol/L for total cholesterol, -0.2 mmol/L for LDL cholesterol, and -0.1 mmol/Lfor HDL cholesterol, indicating lipids and lipoproteins only change minimally in response to normal food intake[1]. On the other hand, since some necessary tests for some pregnant women require fasting, we thus ask fasting blood tests for all participants, even if not all tests require fasting, in order to achieve uniformity.

Based on your kind suggestion, we have added the definition of gestational age in the Materials and Methods as follows:

“Gestational weeks were obtained using the last menstrual period data or the first accurate ultrasound examination if the menstruation was irregular[19].” (Page 3, Line 112-113)

According to clinical practice in Chinese populations, the cut-off value for macrosomia is usually 4kg, as documented on the Maternal and Child Health Guidelines, and almost all studies based on Chinese populations utilize this cut-off[2-5].

References

[1] Langsted A, Nordestgaard BG. Nonfasting versus fasting lipid profile for cardiovascular risk prediction. Pathology. 2019;51(2):131-141. doi:10.1016/j.pathol.2018.09.062

[2] Pan XF, Tang L, Lee AH, et al. Association between fetal macrosomia and risk of obesity in children under 3 years in Western China: a cohort study. World J Pediatr. 2019;15(2):153-160. doi:10.1007/s12519-018-0218-7

[3] Yang H, He B, Yallampalli C, Gao H. Fetal macrosomia in a Hispanic/Latinx predominant cohort and altered expressions of genes related to placental lipid transport and metabolism. Int J Obes (Lond). 2020;44(8):1743-1752. doi:10.1038/s41366-020-0610-y

[4] Lei F, Zhang L, Shen Y, et al. Association between parity and macrosomia in Shaanxi Province of Northwest China. Ital J Pediatr. 2020;46(1):24. Published 2020 Feb 18. doi:10.1186/s13052-020-0784-x

[5] Li G, Xing Y, Wang G, et al. Differential effect of pre-pregnancy low BMI on fetal macrosomia: a population-based cohort study. BMC Med. 2021;19(1):175. Published 2021 Aug 4. doi:10.1186/s12916-021-02046-w

Comment 5: 162 Table 1.  Please analyse the fasting and 2hr [glucose] and [TG] as continuous variables.  Our cutoffs for abnormal are arbitrary.  This could be done in addition to the method in the paper

211 percent of what?  Sorry These do not add to 100% even with non GDM, Normal low TG, macrosomia accounted for

Positive predictive value of BMI >28,  TG>what value 

Do you have data on actual outcomes eg shoulder dystocia

Response 5: We sincerely thank you for this comment. GDM was diagnosed using cut-off values determined by the International Association of Diabetes and Pregnancy Study Group: 5.1 mmol/L for fasting serum glucose, 10.0 mmol/L for 1-hour serum glucose, or 8.5 mmol/L for 2-hour serum glucose[1]. The cut-off value for the diagnosis of GDM was fixed. In addition, based on previous studies[2-3], the cut-off point for mTG was considered to be the 90th percentile, and the cut-off value in this study was 5.67 mmol/L. The definition of a "high mTG level" was a value equal to or greater than the 90th percentile; conversely, a "low mTG level" was a value below the 90th percentile.

Table 2 shows the prevalence of GDM, high mTG levels and fetal macrosomia and their 95% confidence intervals across maternal pre-pregnancy BMI categories including normal weight, overweight and obesity. Since data on underweight pregnancies were not included in further mediation analyses, we did not analyze the prevalence of GDM, high mTG levels and fetal macrosomia in the underweight group. However, to avoid unnecessary misunderstandings, we have added the underweight group in Table 2 and revised the corresponding words in the Results as follows.

“The prevalence of GDM, high mTG levels, and macrosomia based on each BMI group was shown in Table 2. Women with obesity prior to pregnancy had a higher prevalence of GDM and high mTG levels (31.4% and 14.7%, respectively), compared to overweight women (23.7% and 12.9%, respectively), women with normal BMI (15.4% and 10.1%, respectively) and underweight women (9.0% and 7.7%, respectively). Likewise, the prevalence of macrosomia in babies born to obese women was the highest (15.6%), higher than in babies born to overweight women (5.9%),  women with normal BMI (3.9%) and underweight women (2.8%).” (Page 6, Line 211-216)

In addition, we were sorry that we did not collect further data on the adverse consequence of being affected by macrosomia like shoulder dystocia.

References

[1] Metzger BE, Gabbe SG, Persson B, Buchanan TA, Catalano PA, Damm P, Dyer AR, Leiva A, Hod M, Kitzmiler JL et al: International association of diabetes and pregnancy study groups recommendations on the diagnosis and classification of hyperglycemia in pregnancy. Diabetes Care 2010, 33(3):676-682.

[2] Jiang XF, Wang H, Wu DD, Zhang JL, Gao L, Chen L, Zhang J, Fan JX: The Impact of Gestational Weight Gain on the Risks of Adverse Maternal and Infant Outcomes among Normal BMI Women with High Triglyceride Levels during Early Pregnancy. 2021, 13(10).

[3] Lin XH, Wu DD, Li C, Xu YJ, Gao L, Lass G, Zhang J, Tian S, Ivanova D, Tang L et al: Maternal High Triglyceride Levels During Early Pregnancy and Risk of Preterm Delivery: A Retrospective Cohort Study. J Clin Endocrinol Metab 2019, 104(4):1249-1258.

Comment 6: 312 sorry I don’t understand the the English. Presumably they are stating It was thought that Glucose was the…

Some have suggested that Maternal insulin resistance is important for maternal hyperglycaemia (with increased amino acid flux to fetus driving fetal hyperinsulinaemia in addition to glucose which drives fatal hyperinsulinaemia AND contributes to the calories) and hypertriglycerideaemia (driving calories to fetus with glucose)

Both glucose and TG concentrations appear important

Response 6: We sincerely thank you for this comment. We have revised the sentence in the original Line 312 as follows:

“It was thought that glucose was the primary source of energy for fetal growth, and altered maternal glucose homeostasis was a well-established risk factor for excessive fetal growth.” (Page 9, Line 321-322)

We completely agree with you that both glucose and TG concentrations appear important. We have included in the Discussion the words that express this view, citing your recommended literature as follows:

“The influence of maternal lipids on birth weight was regarded as crucial as the well-established effect of glucose[46].” (Page 9, Line 353-354)